# Metabolic Disposition and Elimination of Tritum-Labeled Sulfamethoxazole in Pigs, Chickens and Rats

**DOI:** 10.3390/metabo13010057

**Published:** 2022-12-30

**Authors:** Jingchao Guo, Yaqi Sun, Yongxia Zhao, Lingli Huang, Dapeng Peng, Haihong Hao, Yanfei Tao, Dongmei Chen, Guyue Cheng, Xu Wang, Yuanhu Pan

**Affiliations:** 1National Reference Laboratory of Veterinary Drug Residues (HZAU) and MAO Key Laboratory for Detection of Veterinary Drug Residues, Huazhong Agricultural University, Wuhan 430070, China; 2Shandong Academy of Pharmaceutical Sciences, Jinan 250101, China; 3MAO Laboratory for Risk Assessment of Quality and Safety of Livestock and Poultry Products, Huazhong Agricultural University, Wuhan 430070, China

**Keywords:** [^3^H]-sulfamethoxazole, tissue distribution, metabolism, elimination, residual markers, residual target tissue

## Abstract

Sulfamethoxazole (SMZ), as a sulfa antibiotic, is often used in the treatment of various infectious diseases in animal husbandry. At present, SMZ still has many unresolved problems in the material balance, metabolic pathways, and residual target tissues in food animals. Therefore, in order to solve these problems, the metabolism, distribution, and elimination of SMZ is investigated in pigs, chickens, and rats by radioactive tracing methods, and the residue marker and target tissue of SMZ in food animals were determined, providing a reliable basis for food safety. After a single administration of [^3^H]-SMZ (rats and pigs by intramuscular injection and chickens by oral gavage), the total radioactivity was rapidly excreted, with more than 93% of the dose excreted within 14 days in the three species. Pigs and rats had more than 75% of the administered volume recovered by urine. After 7 days of continuous administration, within the first 6 h, radioactivity was found in almost all tissues. The highest radioactivity and longest persistence in pigs was in the liver, while in chickens it was in the liver and kidneys, most of which was removed within 14 days. A total of six, three and three metabolites were found in chickens, rats and pigs, respectively. N_4_-acetyl-sulfamethoxazole (S1) was the main metabolite of SMZ in rats, pigs and chickens. The radioactive substance with the longest elimination half-life is sulfamethoxazole (S0), so S0 was suggested to be the marker residue in pigs and chickens.

## 1. Introduction

Sulfa drugs (SA) are N-substituted derivatives of sulfanilic acid, with stable properties, low price, and strong antibacterial properties. They are one of the main antibacterial drugs used by humans and animals [1]. Sulfamethoxazole (SMZ, Figure 1), 4-amino-N-(5-methyl-3-isoxazolyl) benzenesulphonamide or C_10_H_11_N_3_O_3_S, is a medium-effect sulphonamide and is one of the most widely used sulphonamide antimicrobials [2]. It is mainly used to treat urinary tract, respiratory tract and intestinal infections, as well as being used as a growth promoter and to improve feed efficiency at the sub-therapeutic level in animal husbandry [3]. Nevertheless, it can also cause some adverse reactions, such as allergic reaction [4], nausea, vomiting, hematological toxicity [5], liver toxicity [6,7], kidney damage and so on [8]; in addition, sulfamethoxazole can exist in the ecological environment for a long time [9], causing harm to human life and health through the food chain [10]. Therefore, the safety monitoring of SMZ in edible animals is vital to human health.

Although there have been many reports on the metabolism of SMZ in animals, including the metabolic pathways and main metabolites of SMZ in different species of animals, the material balance and tissue distribution of the drug in different species of animals are still unclear. Studies have reported that the known metabolic pathways of SMZ include the acetylation and oxidation of the N_4_ nitrogen atom, the glucuronidation of the N_1_ atom, and the hydroxylation of the C_5_ methyl group [11,12]. Among them, the metabolic pathway of SMZ in humans, cats, pigs and ruminants is mainly acetylation, while in dogs [13], homing pigeons, snails and tortoises, hydroxylation plays a leading role, and there are metabolic differences in species [14]. Mengelers and Nouws et al. found that the main metabolic pathway of SMZ in pigs is acetylation, but hydroxyl and glucuronic acid derivatives cannot be detected in pig plasma and urine [15]. In addition, these studies did not conduct material balance tests, and did not have specific excretion or tissue distribution data. Therefore, so far we are still not sure whether there are only these few metabolites in pigs, and it is impossible to comprehensively evaluate the metabolism of the drug in pigs and its potential hazards and safety issues [16]. There are literature reports on the metabolism and distribution of SMZ in rats, but the method used in this article is the Bratton-Marshall method, which is not accurate enough and is no longer used. It cannot fully explain the metabolism of SMZ in rats [17]. In addition, with regard to the application of SMZ in chickens, only pharmacokinetic data has been reported, and its metabolism or specific tissue residues have not been reported [18]. The tissue distribution data of drugs in different species is very important for its application. Without tissue distribution data, the residual target tissue cannot be determined. At present, the existing conventional research methods cannot accurately qualitatively and quantitatively analyze the primary and secondary metabolites in the body [19]. Therefore, it is very important to use radiolabeled drugs to study their metabolism and distribution in different species of animals [20].

Therefore, a comprehensive understanding of SMZ’s metabolic pathways, disposal kinetics and excretion characteristics is of great significance for assessing the potential hazards of the compound and food safety. This study conducted a radioisotope tracing study in accordance with the guidelines of the veterinary drug product registration technical requirements coordination international cooperation (2010) [20]. In this study, we compared the metabolite profiles of SMZ in rats, pigs and chickens (qualitative and quantitative). In addition, the excretion rate and route (mass balance of excretion) of [^3^H]-SMZ radioactivity in urine and feces and the distribution of SMZ in tissues after repeated administration of [^3^H]-SMZ were studied. The above research will help us deepen our understanding of the relationship between the toxicity and metabolic disposal of SMZ, and better evaluate the food safety of the compound.

## 2. Materials and Methods

### 2.1. Chemicals

SMZ (>98%) was provided by China Institute of Veterinary Drug Control (Beijing, China). Palladium on Carbon (Pd/C) was obtained from Sigma-Aldrich (Shanghai, China). Reagents such as H_2_O_2_(30%), sodium chloride (Nacl) and anhydrous sodium sulfate (Na_2_SO_4_) were purchased from Sinopharm Chemical Reagent Co., LTD. (Shanghai, China). Solvable (tissue solubilizing fluid), and Ultima Gold scintillation fluid were obtained from PerkinElmer Life and Analytical Sciences (Groningen, The Netherlands; Waltham, MA, USA, respectively). Stop Flow AD scintillation liquid was purchased from AIM Research Co. (Hockessin, DE, USA). Water was purified by using a Milli-Q system (Bedford, MA, USA). All reagents and other chemicals are analytical grade or higher purity.

### 2.2. Tritum-Labeled SMZ

The chemical structure of SMZ as shown in Figure 1. [^3^H]-SMZ was prepared as described briefly in Figure 2. Step (1): 3-bromoaniline 7.23 g (0.04 mol) and 6.7 mL (0.05 mol) triethylamine were dissolved in 100 mL dichloromethane, 10.6 g trifluoroacetic anhydride (TFAH) (0.05 mol) was added drop by drop at 0 °C. Product 1 was formed by the above reaction mixture at room temperature. Step (2): take product 114.0 g (0.015 mol), Na_2_SO_4_ 4.3 g (0.03 mol), slowly added 4 mL (0.06 mol) of chlorosulfonic acid (ClSO_2_OH) and thionyl chloride (SOCl_2_) 1.1 mL (0.015 mol) at 0 °C. The reaction solution was stirred at 30–50 °C for 12 h to produce product 2. Step (3): the product of the previous step and 0.84 g (0.009 mol) 3-amino-5-methylisoxazole were dissolved in 12.5 mL acetonitrile, 0.42 g (0.005 mol) pyridine was slowly added mixture at 0 °C. Then the reactants were reacted at room temperature for 12–24 h to produce product 3. Step (4): product 3 was dissolved in 5 mL 2 moles per liter (mol/L) of aminomethanol solution, stirred at room temperature for 12 h to obtain 2-bromosulfamethoxazole. Step (5): take 2-bromosulfamethoxazole 50 mg and sodium hydroxide 6 mg was dissolved in 5 mL methanol and add 25 mg 10% Pd/C. The above reaction mixture was injected with 250 mmHg tritium gas and reacted at 30 °C for 2 h. After the reaction was completed, a 2695 high-performance liquid chromatography (HPLC) (Waters, Milford, MA) with a 2996 detector was used for purification. The reaction product was eluted with acetonitrile/phosphate buffer (30:70, *v*:*v*), the flow rate was set to 1.0 mL/min, and the temperature was 30 °C, UV detection wavelength was 270 nm; the test time was 25 min. The components corresponding to the retention time of SMZ were collected, the solvent was fully removed, and we obtained a [^3^H]-SMZ with high chemical purity (≥99%), high radio-chemical purity (≥98%), and high specific activity (52.9 Ci/g). The tritium-labeled [^3^H]-SMZ solution remained stable at −20 °C for 6 months with a tritium exchange risk of less than 5%.

Using meta-bromoaniline as raw material, the amino group was protected by trifluoroacetic anhydride and then reacted with chlorosulfonic acid to obtain 2-bromo-4-trifluoroacetylbenzenesulfonyl chloride. Then, it was condensed with 3-amino-5-methylisoxazole, and after alkaline hydrolysis and deprotection, 2-bromo-sulfamethoxazole was prepared, and it was reacted with tritium gas under the action of palladium-carbon catalyst and alkali acceptor to generate tritium-halogen exchange to produce 2-[^3^H]-sulfamethoxazole.

### 2.3. [^3^H]-SMZ Injection

For rat and pig experiments, the non-labeled SMZ was weighed accurately. Then, 5 moles per liter of propylene glycol, 0.9 moles per liter of triethanolamine, and a certain amount of [^3^H]-SMZ stock solution were added, and then diluted to a certain specific activity (0.1 Ci/g) with water, which was used in rat and pig experiments.

### 2.4. [^3^H]-SMZ Oral Solution

Unlabeled SMZ and [^3^H]-SMZ were accurately weighed and dissolved in 0.5% carboxymethyl cellulose to make an aqueous solution. The specific activity of [^3^H]-SMZ was 0.1 Ci/g, which was used in chicken experiments.

### 2.5. Animals

In this experiment, pathogen-free male and female Wistar rats (8 weeks old), weighing 200 ± 20 g, were obtained from the Experimental Animal Research Center of Huazhong Agricultural University (Wuhan, China). Healthy castrated hybrid (Duroc ×Dabai) pig (60 d), weighing 25 ± 2 kg, were obtained from Yongsheng Animal Husbandry Co., Ltd. (Yingcheng City, China). Yellow-feathered broilers (70 d, 1.5 ± 0.2 kg) were purchased from Xianning Wens broiler chicken plant Co., Ltd. (Xianning, China). One week before the experiment, all the animals were offered basal feeds without antimicrobial agent compounds. The animals were kept in the animal room, the temperature was 20–26 °C, the relative humidity of the room was 40–70% and a 12 h light/dark cycle. During the adaptation period, all animals were offered water freely, carry out standard feeding and management. The animals were kept on fasting for 12 h before administration of antimicrobial agents/compounds. All experimental procedures and protocols are in accordance with International Cooperation on Harmonisation of Technical Requirements for Registration of Veterinary Medicinal Products (VICH) guidelines GL.46 (2012) and Food and Drug Administration (FDA, Washington, DC, USA) Good Laboratory Practices for Non-Clinical Laboratory Research [20]. This study was approved by the Animal Ethics Committee of the Faculty of Veterinary Medicine (Huazhong Agriculture University, Wuhan, China) and the approval code is HZAURA-2019-0001.

### 2.6. Animal Experimental Design

#### 2.6.1. Excretion and Metabolism Studies

Six Wistar rats (three females and three males), six chickens and four pigs were housed in clean metabolic cages. Pigs and rats were weighed and single intramuscular injected with [^3^H]-SMZ injection at 25 mg/kg b.w. (0.1 Ci/g). The broiler chickens were given the same dose by oral gavage. According to the recommended guidelines, the urine and feces samples of these animals were collected at different time intervals. The remaining samples of the cage were stored together with the collected excrement. At the end of the experiment, all animals were anesthetized and sacrificed in accordance with euthanasia guidelines (AVMA, 2001). Urine and feces samples were stored at −80 °C.

#### 2.6.2. Distribution and Residue Studies

The pigs (20) and chickens (30) were randomly assigned to five equal treatment groups, and [^3^H]-SMZ was given by intramuscular injection (chickens were given by oral gavage) twice a day for 7 consecutive days at 25 mg/kg body weight. The rats (36, half males and half females) were divided into six groups and given a single intramuscular injection of 25 mg/kg b.w. after administration for 0.5, 3, 7, 14, 28 d (rats 1 h, 2 h, 2 h, 6 h, 24 h, 24 h, 3 d, 7 d; chicken 0.25, 3, 7, 14, 28 d), the three animals were randomly selected for the slaughtering, respectively. The guidelines were followed for metabolism and residual kinetics (VICH, 2011) to collect samples, including heart, liver, spleen, lung, kidney, adrenal glands, muscle, fat, skin, brain, gonads, gastrointestinal tract and bladder, and they were stored at −80 °C until analysis. In addition, the animals in the blank group (pigs and chickens) and the animals in the treatment group were given physiological saline and 0.5% carboxymethyl cellulose aqueous solution in exactly the same dose and in exactly the same way. Another 6 rats were administered physiological saline at the same dose and in exactly the same way as the other animals. The same as above, at each sampling time point, a non-administered animal was killed and a sample was taken to obtain a blank sample. The medication groups samples and blank samples were washed and their weight was recorded. Afterwards, all samples were homogenized and divided into two parts. One part was used to detect the total radioactivity, and the other part was used for SMZ and metabolite analysis. In the process of sample preparation and analysis, the blood and bile preparation methods were followed (As shown in Section 2.7.1.). The procedure was the same as the urine sample preparation methods, and the organ and tissue preparation methods were the same as the feces sample preparation methods.

#### 2.6.3. Sample Analysis

The total radioactivity in the samples were measured by Packard Tricarb 2900 liquid scintillation analyzer (LSC). The pretreated 200 μL liquid sample (urine, blood, and bile) was taken and 10 mL of Ultima Gold scintillation fluid was added for total radioactivity detection by LSC. Then, 200 mg of solid samples (feces and tissues) were accurately weighed, and 2 mL of digestion solution were added and mixed well, then the samples were placed in a 55 °C water bath for 6 h, and cooled to room temperature for later use. Then, 0.1 mol/L disodium edetate (EDTA-2Na) and 30% H_2_O_2_ of 200 μL were added and mixed evenly. At room temperature, 10 mL Ultima Gold scintillation fluid was added, and radioactivity determination was performed after standing [21].

### 2.7. Quantification and Structural Identification of Metabolites

#### 2.7.1. Urine, Blood and Bile Preparation

The weight of each aliquot was 0.1% of the total fraction weight (0.25% for urine). Each sample was centrifuged at 4 °C for 5 min at 4000 rpm/min. The supernatant was collected and 2 mol/L methanol-mixed acid (methanol: 0.5% perchloric acid, 8:1, *v*:*v*) was added. Then, the mixture was centrifugated at 10,000 r/min for 10 min, the supernatant was collected and dried with nitrogen at 45 °C. The residues were dissolved with 0.5 mL of mobile phase and mixed for 30 s. The mixture was subjected to Liquid Chromatography tandem on-line isotope detector/Time of Flight Mass Spectrometry (LC-ν.ARC/MS-IT-TOF) system analysis.

#### 2.7.2. Feces or Tissues Preparation

The feces and tissues (2 g) were extracted with 10 mL methanol and water (4:1, *v*:*v*). The mixture was vortexed and sonicated for 5 min and 10 min, respectively, and finally the samples were centrifugated at 8000 rpm/min for 10 min. The supernatants were transferred into a new centrifuge tube and the residues were further extracted with 10 mL methanol and water (4:1, *v*:*v*). The supernatants were combined and evaporated to dryness under a nitrogen stream at 45 °C. The residues were dissolved with 0.5 mL of mobile phase and vortexed mixed for 30 s. Then, 1 mL n-hexane was added to remove the lipid, the mixture was vortexed evenly mixed, centrifugated at 10,000 rpm/min for 10 min and the n-hexane layer was discarded. The mixture was prepared for LC-ν. ARC/MS-IT-TOF system analysis. The SMZ and its metabolites in the prepared samples were determined by a ν. ARC radiation detector and a hybrid IT/TOF mass spectrometer connected to the HPLC system (Shimadzu Corp., Kyoto, Japan). The HPLC system was equipped with a solvent delivery pump (LC20AD), autosampler (SIL20AC), DGU20A3 degasser, photodiode array detector (SPDM20A), communication base module (CBM20A), and a column oven (CTO20AC). Separation was performed on a X Bridge C18 column (150 × 2.1 mm; 3.5 μm) using a gradient elution consisting of mobile phase A (0.1% formic acid in water) and mobile phase B (acetonitrile). The gradient was as follows: 0–5 min, 12% B; 5–10 min, 12–16% B; 10–20 min, 16% B; 20–30 min, 16–50% B; 35 min, 12% B; 35.01–40 min, 12% B. The injection volume was 10 μL, and the flow rate was 0.2 mL/min. Photo-diode array (PDA) detection was performed from 200 to 400 nm. The sample chamber in the autosampler was maintained at 4 °C, whereas the column was set at 25 °C. The total effluent from the detector was transferred directly to the hybrid IT-TOF mass spectrometer without splitting. Metabolites were isolated and identified using a Shimadzu high-performance liquid chromatography-mass spectrometry-ion trap-time-of-flight (LC-MS-IT-TOF) hybrid mass spectrometer equipped with an electrospray ionization (ESI) source and operated in the positive mode. The eluate from the chromatographic separation was divided into two portions. The portion for metabolic quantification was applied to a ν.ARC 2.0, and the rest was analyzed and identified by ion trap time-of-flight tandem mass spectrometry (IT-TOF-MS/MS). Radiochemical detection was performed using the ν.ARC 2.0 Radio-LC system Detector RS232 equipped with a 500 μL detector cell (AIM Research Co., Hockessin, DE, USA). The instrument was operated in the dynamic liquid scintillation flow mode, and the HPLC eluent was mixed 1/5 with Stop Flow AD scintillation liquid. After radiochemical detection, the total effluent from the detector was transferred directly to the hybrid IT/TOF mass spectrometer without splitting.

The IT-TOF-MS-MS instrument was equipped with an electrospray ionization (ESI) source operated in the positive ionization mode. Mass spectroscopic analyses were carried out on a full-scan mass spectrometer with a mass range of 100−500 Da. Liquid nitrogen was used as the nebulizing gas at a flow rate of 1.5 L/min. The interface and detector voltages were set at 4.5 and 1.6 kV, respectively. The curved desorption line (CDL) and heat block temperatures were both 200 °C. The MS2 spectra were produced by collision-induced dissociation (CID) of the selected precursor ions with argon (Ar) as the collision gas. The ion accumulation time and relative collision energy were set at 50 ms and 50%, respectively. Data acquisition and processing were carried out using LC/MS solution software version 3.41, supplied with the instrument. The mass numbers corresponding to the particular elemental compositions were also calculated by the formula predictor and generated more than one formula proposed by the software. Therefore, an accuracy error threshold of 5 mDa was set as the limit for the calculation of possible elemental compositions.

#### 2.7.3. Validation of the Quantitation Methods

All the samples were quantitated by LSC. The limit of quantification (LOQ) of LSC was detected by determining a series of [^3^H]-SMZ working solutions with different radioactivity. The lowest concentration resulting in a deviation of 5% was designated the LOQ. The LOQ of [^3^H]-SMZ in different samples was 13.5 μg/kg. Two microliters of [^3^H]-SMZ stock solution were diluted 10^10^ times with methanol, and then, 20 μL, 40 μL and 80 μL of diluted [^3^H]-SMZ was added to the blank pig urine, feces, blood and tissues, respectively. The samples were prepared as described above, followed by the determination of the radioactivity of the extracted samples with LSC and LC/LSC. All radioassays were performed in triplicate. On comparison of the extracted radioactivity with the total radioactivity of the diluted solution, the accuracy of [^3^H]-SMZ quantitation in the different samples ranged from 82.4 to 103.2%, with a relative standard deviation of less than 10%. Nine diluted [^3^H]-SMZ stock solutions with different radioactivities were detected by LSC and LC/LSC, and each sample was measured in triplicate. The correlation equation between LSC and LC/LSC was determined to be y = 10.757x + 1229.1 (r = 0.9942). To calculate the content of volatile radioactivity, such as [^3^H]-H_2_O, another duplicate set of samples was freeze-dried and reconstituted in water (“dry samples”) prior to analysis. All radioassays were performed in triplicate. The ^3^H water exchange in the excretions was between 0.2% and 2.4% [22].

#### 2.7.4. Data Analysis

The total concentrations of SMZ-related compounds in various samples were calculated by the following formula: A = A1m1/(RM), where A is the total concentration of SMZ-related compounds in the samples (μg/kg), A1 is the total radioactivity of the samples (dpm), m1 is the total drug mass for each animal (μg), R is the total radioactivity dosed to each animal (dpm), and M is the mass of the samples (kg). The concentrations of radioactive substances were quantitated by reversed phase HPLC with radiochemical detection. The linear relationship between the cpm (X), which was obtained from the νARC detector, and dpm (Y) from Tricarb 2900TR was y = 10.757x + 1229.1, (r = 0.9942). The total residue concentrations and concentrations of the main metabolites were subjected to logarithmic transformation and regressed with the time. The depletion equation and the half-life (t_1/2_) of the total residues and main metabolites were described as C = C_0_e^−kt^ and t_1/2_ = 0.693/k (C: radioactive metabolite concentration at any time point, C_0_: initial concentration. K: elimination rate constant), respectively. The ratio of the identified metabolite with the total residues was calculated as follows:X = AUC_M_/AUC_T_ × 100%

X is the percentage of the individual radioactive substances within the total radioactive residues. AUC_M_ is the area under the curve of the individual radioactive substances in the specific samples. AUC_T_ is the area under the curve of the total radioactive residues in the specific samples.

## 3. Results

### 3.1. Radioactivity Excretion of SMZ in Animals

The mean cumulative recovery of [^3^H]-SMZ in the feces and urine of pigs, chickens and rats after a single dose/ injection was summarized in Table 1. The results revealed that the peak of total radioactivity recovery in feces appeared at 0–0.5 d after drug withdrawal, in which the total radioactive recovery rate of pigs, chickens and rats exceeds 65%, 60% and 59%, respectively. In one day, the cumulative excretion recovery rates of the three animals reached 83%, 75% and 83%, respectively. Three species excreted more than 88–92% of a dose in 3 d. Chickens had a slower excretion rate than pigs and rats. In pigs and rats, respectively, more than 80% and 75% of SMZ was excreted through urine, whereas 14.7 and 21.4% was excreted through feces, indicating that SMZ was well absorbed and was mainly excreted through the kidneys. Within 14 d, the radioactive recovery rates of SMZ excretion in male and female rats were 98 and 97%, indicating that the excretion of SMZ in male and female animals were similar, i.e., there was no obvious difference.

### 3.2. Metabolism of SMZ in Animals

In rats, pigs and chickens, five metabolites were found through the ARC system and confirmed through LC-MS-IT-TOF system. In this study, we further identified these five metabolites by comparison with the parent drug’s molecular mass change, accurate mass measurement and chromatographic retention time. N_4_-acetyl sulfamethoxazole (S1) and N_4_-hydroxyamino sulfamethoxazole (S3) were synthesized to determine their structures. Table 2 lists the possible metabolite structures of SMZ and the corresponding spectral data. The percentage of a single metabolite in the total concentration of SMZ-related compounds within 12 h after administration is shown in Table 3.

In rats, within 0.5 d after intramuscular injection, only S0 and S1 were detected in urine, which accounted for 48.4 and 51.6%. In addition, trace amounts of the glucuronide conjugation metabolite N_4_-glucuronidsulfamethoxazole (S5) were detected in rat urine by LC-MS-IT-TOF. In feces, only S0 and S1 were detected at 0–14 d after administration (Table 3). At 0–0.5 d, S0 and S1 accounted for 38.3 and 61.7% of the total fecal residues, respectively (Table 3). At the same time, S0 and S1 were also detected in liver, kidney, muscle, fat and blood within 6 h after administration (Table 4).

In pigs, SMZ metabolism was the same as in rats. S0 and S1 were detected in the urine after administration by LC-ν.ARC (Liquid Chromatography tandem On-line isotope detector) in Figure 3. There were trace amounts of S5 first detected in pig urine at 0–0.5 d by LC/MS-IT-TOF as shown in Figure 3. The present drug (S0) and S1 accounted for 24.6% and 75.4% of the total radioactivity in the urine from 0–0.5 d, respectively (Table 3). In the feces, only S0 and S1 were detected at 12 h post-dosing, with ratios of 12.1% and 87.9%, respectively (Table 3). S0 was not detected after 7 d of administration, only S1 was detected. After 12 h of administration, only S0 and S1 were detected in the feces samples, accounting for 12.1% and 87.9%, respectively (Table 3), S0 was not detected 7 d after administration. S0 and S1 were found in the blood and bile after 12 h of the final dosing, but only S0 and S1, not S5, were observed in the liver, kidney, muscle and fat (Table 4).

In chickens, five metabolites were identified (Figure 4). Four metabolites were detected in chicken feces at 12 h postdosing by LC-ν.ARC. In addition to S0 and S1, two radioactive compounds were detected, 5-hydroxysulfamethoxazole (S2) and N_4_-hydroxysulfameth-oxazole (S3). S0, S1, S2 and S3 accounted for 69.6, 16.4, 9.6 and 4.4% of the total radioactivity, respectively (Table 3). A trace amount of metabolite S5 was also detected in chicken feces at 12 h by LC-MS-IT-TOF as shown in Figure 4A. At 0.5–1 d after administration, S3 was no longer detected in chicken feces, but a new metabolite N_4_-sulfatesulfamethoxazole (S4) was found, which accounted 2.06% of the total radioactivity as shown in Figure 4B. At 1–14 d after administration, only S0 and S1 were detected in chicken feces. S2, S3 and S4 were metabolites specific to chicken. In the blood, the parent drug and three metabolites (S1, S2 and S3) were observed at 6 h postdosing, in proportions of 91.2%, 6.7%, 1.2% and 0.9%, respectively (Table 3). Whereas S3 was not detected in bile, only S0, S1 and S2 were found at 12 h, which accounted for 88.2, 9.3 and 2.5 of the total radioactivity, respectively (Table 3). S0 was the major metabolite in blood and bile.

Overall, SMZ has the largest number of metabolites observed in chickens, and its metabolism is more extensive than in rats and pigs. S0, S1 and S5 were found in rats, chickens and pigs, and S1 was the major metabolite in three species. S2, S3 and S4 were only observed in chickens. These interspecies differences can be explained by a difference in the amount and type of iso-enzymes in the cytochrome P450 system in the liver [23]. According to the structure, persistence and percentage of metabolites, the possible metabolic schemes of SMZ in pigs, chickens and rats were shown in Figure 5.

### 3.3. Distribution

The tissue distribution of SMZ in chickens, pigs and rats is similar. The tissue distribution of [^3^H]-SMZ in the three animals is shown in Figure 6. SMZ-derived substances could be rapidly and widely distributed in various tissues and fluids of animals. After 7 consecutive days (single-dose rats) intramuscular injection (gavage in chickens) [^3^H]-SMZ, the radioactivity was widely distributed and could be observed in almost all collected tissues. The highest levels of radioactivity of all the collected substrates in rats, chickens and pigs were detected at 1, 6 and 12 h time points, respectively. The total radioactive residues in blood, kidney and bile were the highest compared with other tissues, and the second highest levels were in the hearts, livers, lungs, chicken craws, spleens, stomachs, intestines, skins, muscles and brains, whereas the lowest level was found in the fat. There were also some differences in the distribution of SMZ among the three animals. The radioactivity in pig muscles and skin were lower than that in chickens and rats; this might be due to species differences or difference in the modes of drug administration.

X axis: the main tissues of [^3^H]-SMZ distribution, Y axis: the radioactivity concentration of [^3^H]-SMZ (μg/kg), among which, rats lists the [^3^H]-SMZ tissue distribution results at seven time points, pigs and chickens list [^3^H]-SMZ tissue distribution results at five time points.

### 3.4. Residue Depletion 

In pigs, S0 and S1 were found in the liver, kidney, muscle and fat (Table 4). In the liver, kidney, muscle and fat of pigs, S0 accounts for 34.4%, 33.5%, 55.9% and 41.9% of the total residues within 12 h after administration, respectively. It was observed that main residual metabolite of SMZ was S1, which accounted for 65.6%, 66.5%, 44.1% and 58.1% of the total residual amount of four tissues at the second time point, respectively. The drug-related radioactivity declined rapidly within 3 d after administration. On the 28th day after administration, radioactivity was still present in the liver, muscle, bile and blood. After statistics, it was found that the t_1/2_ of total residues in liver, kidney, muscle and fat were 5.76 d, 4.94 d, 3.84 d and 5.80 d, respectively. It demonstrated that that liver and fat were the main storage tissues that eliminated the SMZ and its metabolites slowly in pigs. In addition, the liver contained most of the residues, and concentration was 26 μg/kg at 28 d. Thus, the liver was proposed as the target tissue of SMZ in pigs. S0 was the slowest component to be eliminated. In pig liver, the elimination half-life of S0 (t_1/2_ 5.63 d) was essentially the same as that of the total residues (t_1/2_ 5.76 d), and the depletion curve was similar. Therefore, S0 was proposed as the residual marker residue of SMZ in pigs.

In chickens, S0–S4 could be detected simultaneously in chicken kidney tissue at the first time point. S1 was not detected at 7 days, S2, S3, and S4 were not detected at 3 days after administration. S0 and S1 were common metabolites in chicken liver, muscle and fat (Table 5). As the most important metabolite, the residual amount of S1 in the four tissues of chickens accounted for 71.7%, 1.0%, 42.1% and 70.0%, respectively, 6 h after administration. At 28 d after administration, radioactivity could still be detected in almost all tissues. The elimination of total residues in liver, muscle, and fat was relatively slow, with elimination half-lives of 5.16 d, 3.81 d, 5.16 d, and 5.36 d, respectively (Table 6). The residual radioactivity in muscle and liver was relatively high, ranging from 50 to 60 μg/kg. At the same time, it is not observed in the fat. Therefore, liver and muscle were identified as the target tissues for SMZ residues in chickens. S0 was the slowest elimination component, with t_1/2_ of 4.90 d and 5.29 d, respectively, in liver and muscle. Therefore, S0 was defined as the residual marker of SMZ in chickens.

In general, the drug-related radioactivity declined rapidly within 3 d after administration. Thereafter, within 14 d (7 d for rats) after administration of SMZ in these three species, its decline was slow, and it was still detectable in almost all tissues. In rats, SMZ showed the fastest depletion time, with t_1/2_ ranging from 1.24 to 2.56 d, followed by chickens with t_1/2_ ranging from 3.81 to 5.36 d. The pigs revealed the slowest depletion time, with t_1/2_ ranging from 3.48 to 6.48 d. In rats, the total radioactive residual time in the kidney was the longest, and in pigs and chickens, the total radioactive residual time in the fat and liver was the longest. Within 28 d after administration, the total amount of residual radioactivity in liver and muscle was highest in pigs and chickens. S0 and S1 were the major radioactive components in the three species. Other trace metabolites were rapidly depleted and disappeared within 3 d. The slowest to eliminate was S0 compared to other metabolites, with t_1/2_ ranging from 1.39 to 3.74 d, 4.09 to 5.63 d and 4.63 to 5.32 d in rat, pig and chicken tissues, respectively. S1 had t_1/2_ ranging from 0.22 to 0.72 d, 1.15 to 3.78 d and 0.87 to 3.64 d in the tissues of these three species, respectively. Therefore, these results indicate that the residual target tissue of SMZ in pigs may be liver, and the residual target tissues of chicken may be liver and muscle and S0 could be applied as a residual marker in food-producing animals.

## 4. Discussion

In the early stage, there were few studies on the material balance and metabolism of sulfamethoxazole in livestock and poultry, but in chickens, no relevant studies have been reported. Only metabolites in urine and plasma have been structurally identified in pigs, without complete material balance data. Therefore, in order to better elucidate the disposal and metabolism of sulfamethoxazole, this study investigated the metabolism and disposal of SMZ in three species by detecting radioactive labeled SMZ in pigs, chickens and rats [24]. Our results revealed that SMZ was rapidly excreted in these three species. Within 12 h of administration, approximately 60–65% of the total radioactive material was recovered from urine and feces. At 3 d, more than 87% of [^3^H]-SMZ and its metabolites from these three species were recovered. Kitakaze reported that after a single oral administration of SMZ in rats, most of the SMZ was excreted within the first 24 h, and approximately 72–82% was excreted in urine [17]. In this study, 75–80% of a single intramuscular injection of [^3^H]-SMZ in rats was excreted in the urine. Peak excretion occurred at 0–1 d. In pigs, 80% of SMZ and its metabolites were excreted in the urine. The above results indicate that the SMZ was primarily excreted in the urine. The radioactivity of acetyl-sulfamethoxazole accounts for 80% of the total radioactivity in pig urine, and unchanged SMZ accounted for 15%, indicating that acetylation was the main route of excretion. Acetyl-SMZ was excreted by glomerular filtration and active tubule secretion, while SMZ was excreted through passive processes.

SMZ has the fastest excretion rate in rats, followed by pigs and chickens. This may be related to the pH of the excreta and the types of metabolites. The pKa value of SMZ is 6.0, so its degree of ionization in the urine is significantly affected by changing in urinary pH. Previous studies revealed that when the pH value of the urine rises from 5 to 8, the elimination half-life of SMZ in the body is significantly reduced, indicating rapid excretion of alkaline urine and slow excretion of acidic urine [25]. The pH value of rat urine is around 8.5, while that of pig urine is only 6.2. The alkaline urine in rats resulted in increased SMZ ionization, so SMZ was excreted in rats at the fastest rate. The phase II metabolite of SMZ will be excreted at an accelerated rate in the body. Among the three animal excrements, the chicken had the lowest acetylation ratio and the highest prototype ratio, so its excretion rate was slightly slower than that of pigs and rats.

Metabolite analysis show that SMZ has a total of five metabolites in pigs, chickens and rats, which are more widely metabolized in chickens. Five metabolites were detected in chickens, but only two metabolites (S1, S5) were detected in rats and pigs. This shows that there are interspecies differences in the metabolism of SMZ. This mainly depends on the chemical structure of the drug and the composition and content of enzymes in different species [26]. These metabolites include acetylation metabolite S1, two hydroxylation metabolites (S2, S3), sulphate conjugation metabolite S4 and glucuronide conjugation metabolite S5. These metabolites indicated that SMZ was mainly metabolized in three animals through N_4_ acetylation, N_4_ glucuronic acid binding, N_1_ and N_4_ hydroxylation [27].

SMZ can be acetylated by N-acetyltransferase (NAT2), oxidized by CYP2C9 to generate hydroxylamine metabolites [12], and conjugated by glucuronosyltransferase to generate N_1_-glucuronate conjugates. Acetylation is the main metabolic pathway of SMZ in the body, leading to the production of the main metabolite S1, which remains the same in the three species. Hydroxyl and glucosidate derivatives have not been detected in pig plasma and urine by HPLC [16]. However, in this experiment, glucuronide conjugation metabolites were detected in pig urine and blood by LC-MS IT-TOF. The instrument used in this experiment was more accurate.

SMZ metabolism to produce hydroxylamine (S3) is thought to cause adverse reactions after medication. A previous study has shown that hydroxylamine itself is neither highly reactive nor directly cytotoxic but rapidly oxidizes to form the more cytotoxic and reactive nitroso metabolite [28]. In the present study it was shown that S3 is acetylated to N-acetoxy-SMZ by human recombinant NATs. In vitro cytotoxicity test showed that SMZ metabolism to produce N-acetoxy-SMZ is more toxic than S3 [30]. The Ames test showed that N-acetoxy-SMZ can cause weak mutations in the TA100 strain of Salmonella typhimurium [29]. These results indicate that the N-hydroxylation and N-acetoxy pathway of SMZ may be closely associated with its toxicity.

In early studies, the Bratton-Marshall method was used to study the distribution of sulfamethoxazole in rats [17], and the elimination of sulfamethoxazole in chicken tissues by this method was also reported in literature, but comprehensive tissue distribution and residual elimination data were not given [30]. The study on distribution and elimination was only conducted for 4 days, which was far from the requirements of the sulfonamides residue elimination test, and the Bratton-Marshall method had the problem of inaccurate quantitative measurement. In addition, the distribution and residue elimination of sulfamethoxazole in pigs have not been reported. Therefore, in order to solve the above problems, tritium-labeled sulfamethoxazole was used in this study, and its distribution and elimination rules in pigs, chickens and rats were studied by radioactive labeling combined with liquid chromatography-mass spectrometry, so as to obtain more reliable distribution and elimination data. Our research work demonstrated that SMZ was absorbed rapidly and widely distributed in the body. Radioactivity could be detected in almost all organs and tissues in pigs, chickens and rats 6 h after administration. Radioactivity detection revealed that the highest concentration of radioactivity was observed in the serum at each time point of sacrifice. The radioactivity concentration in tissue is always lower than that in serum, and the highest concentration in kidney, liver, lung and bile is significantly higher than other tissues at every time point. The distribution of SMZ in the lungs helped it to prevent and treat pulmonary infections caused by sensitive bacteria Pasteurella multocida and Actinobacillus pleuropneumoniae [31]. The static liquid scintillation instrument detected that there was also radioactivity in the animal’s brain, indicating that SMZ and/or its metabolites can pass through the blood-brain barrier through blood circulation. However, the high-level distribution of radiolabeled drugs and/or their metabolites in certain tissues indicate that they are easily exposed in toxicology studies. When the dose is higher, it will cause obvious organ damage to the animal [32]. After 14 d of administration, high levels of residual radioactivity were still detected in liver and kidney, indicating that liver and kidney are the focus of research on SMZ food safety and toxicology. The half-life of SMZ elimination in the kidney and liver indicates that the elimination time of SMZ in the liver is longer than in other tissues. Kidney and liver are important target organs of SMZ toxicity.

In summary, this work mainly conducted a comparative study on the metabolism, excretion, distribution and elimination of SMZ in pigs, chickens and rats by using radioactive tracing methods, supplemented the previous data on the mass balance, excretion and tissue distribution of SMZ. Furthermore, we confirmed the metabolites of SMZ in different species of animals and we found the main metabolites related to SMZ toxicology and food safety. Studies have found that SMZ is rapidly excreted from animals mainly through urine and is widely distributed, and metabolized in rats, pigs and chickens. N_4_ acetylation is the main biotransformation pathway of SMZ in three species, and it might be related to the toxicity of SMZ and food safety. Since SMZ mostly exists as prototypes in pigs, chickens and rats, S0 is proposed as a residual marker for food safety control. Moreover, SMZ was slowly eliminated in the liver. Therefore, the liver is recommended as a residual target organ for toxicology and food safety issues. Residue elimination of SMZ and its metabolites in food animals are closely related to food safety. The designation of residue markers and target tissues is very important for food safety monitoring of sulfamethoxazole. This study provides comprehensive information on the disposal and tissue depletion of SMZ in animals and identified residual markers and residual target tissues. These results provided important information for the safety and risk assessment of SMZ in food animals, and lay the foundation for a deeper understanding of the relationship between SMZ’s metabolism, elimination and toxicology in the food animals.

## Figures and Tables

**Figure 1 metabolites-13-00057-f001:**
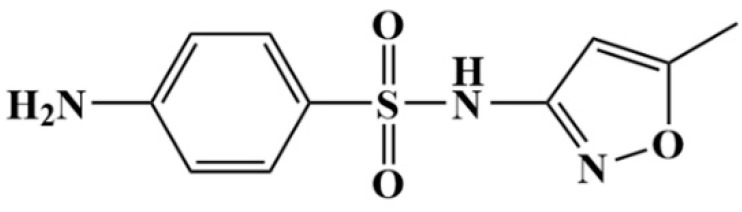
Chemical structure of Sulfamethoxazole.

**Figure 2 metabolites-13-00057-f002:**
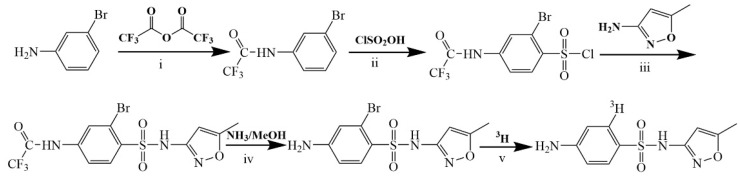
Synthesis scheme of [^3^H]-SMZ.

**Figure 3 metabolites-13-00057-f003:**
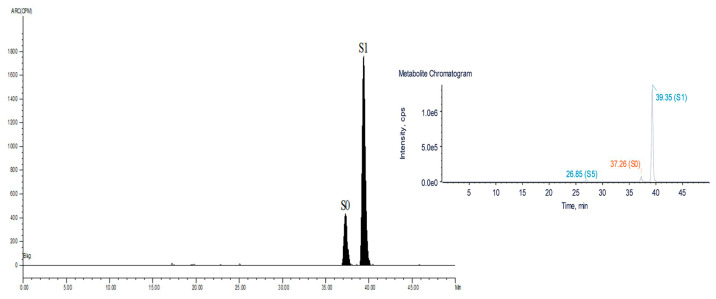
Radiochromatogram and total extracted ion chromatogram (EIC) of SMZ and its metabolites in the urine of pigs at 0–0.5 d. S0, sulfamethoxazole; S1, N4-acetyl-sulfamethoxazole; S5, N4-glucuronidsulfamethoxazole.

**Figure 4 metabolites-13-00057-f004:**
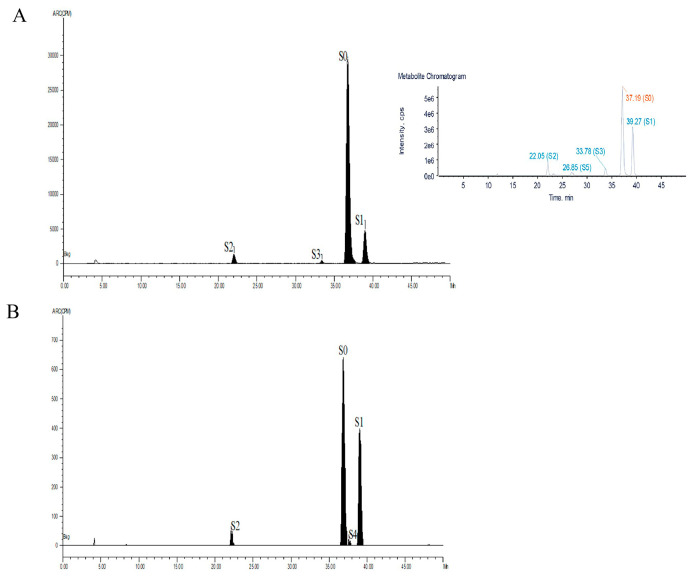
Radiochromatogram and total EIC of SMZ and its metabolites in the feces of chickens at 0–0.5 d (**A**) and 0.5–1 d after administration of [^3^H]-SMZ (**B**). S0, sulfamethoxazole; S1, N4-acetyl-sulfamethoxazole; S2, 5-hydroxysulfamethoxazole; S3, N4-hydroxysulfameth-oxazole; S4, N4-sulfatesulfamethoxazole; S5, N4-glucuronidsulfamethoxazole.

**Figure 5 metabolites-13-00057-f005:**
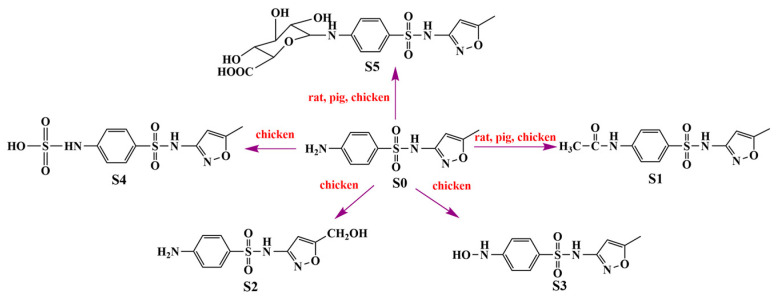
Proposed metabolic pathway of SMZ in animals. Glucuronidation and acetylation are common metabolic pathways in pigs, chickens and rats. Hydroxylation and sulfate conjugation are unique metabolic pathways for chickens.

**Figure 6 metabolites-13-00057-f006:**
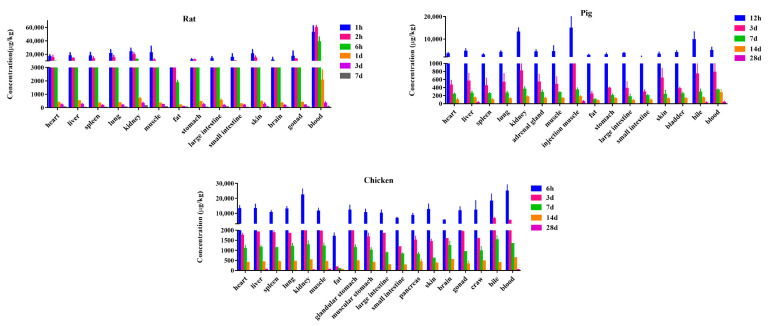
Tissue distribution of [^3^H]-SMZ in rats, pigs and chickens.

**Table 1 metabolites-13-00057-t001:** Recovery of total radioactivity in the urine and feces of pigs, chickens and rats. After administration of [^3^H]-SMZ at 25 mg/kg b.w.

Time (d)	Pigs		Chickens	Rats			
				Male		Female	
	Urine	Feces	Excreta	Urine	Feces	Urine	Feces
0–0.5	60.02 ± 16.54	5.56 ± 0.42	60.65 ± 13.08	45.33 ± 4.79	17.3 ± 1.16	45.38 ± 6.09	14.31 ± 1.73
0.5–1	15.3 ± 8.73	2.69 ± 3.12	15.08 ± 6.87	19.31 ± 7.15	5.3 ± 1.43	19.41 ± 2.15	4.45 ± 1.87
1–3	4.52 ± 2.22	5.19 ± 3.36	11.91 ± 9.67	7.49 ± 2.04	0.35 ± 0.14	10.09 ± 3.14	1.02 ± 0.72
3–7	0.59 ± 0.17	0.82 ± 0.34	5.48 ± 1.89	2.7 ± 1.89	0.25 ± 0.14	3.97 ± 1.24	0.42 ± 0.28
7–14	0.16 ± 0.04	0.45 ± 0.3	0.26 ± 0.13	0.49 ± 0.3	0.05 ± 0.03	0.38 ± 0.17	0.05 ± 0.03
0–14	80.59 ± 5.72	14.72 ± 1.31	93.39 ± 2.28	75.32 ± 4.54	23.24 ± 1.79	77.9 ± 5.93	19.58 ± 2.09
Total	95.31 ± 4.41		93.39 ± 2.28	98.56 ± 2.82		97.3 ± 1.16	

**Table 2 metabolites-13-00057-t002:** Metabolite information of SMZ in pigs, chickens and rats.

Compound	Chemical Name	[M+H] (Da)	Formula	Major Product Ions
S0	4-Amino-N-(5-methyl-3-isoxazolyl)	254.0521	C_10_H_11_N_3_O_3_S	156, 108, 92, 188, 99, 147
S1	N-Acetyl sulfamethoxazole	296.0627	C_12_H_13_N_3_O_4_S	198, 134, 188
S2	4-Amino-N-(5-hydroxymethyl-3-isoxazolyl)	270.0564	C_10_H_11_N_3_O_4_S	252, 204, 156, 147, 108, 92
S3	4-Hydroxyamino-N-(5-methyl-3-isoxazolyl)	270.0564	C_10_H_11_N_3_O_4_S	204, 172, 108
S4	Sulfamethoxazole-N_4_-slufate	334.0172	C_10_H_11_N_3_O_6_S_2_	254, 156
S5	sulfamethoxazole- N_4_-glucuronide	430.0945	C_16_H_19_N_3_O_9_S	332, 254, 156, 147, 99

**Table 3 metabolites-13-00057-t003:** The percentage of the individual metabolites compared with the total concentration of SMZ-related within 12 h after dosing (%).

Compound	Rat			Pig				Chicken		
	Urine	Feces	Plasma	Urine	Feces	Plasma	Bile	Extra	Bile	Plasma
S0	48.4	38.3	94.5	24.6	12.1	80.3	74.5	69.6	88.2	91.2
S1	51.6	61.7	5.5	75.4	87.9	19.7	25.5	16.4	9.3	6.7
S2	ND	ND	ND	ND	ND	ND	ND	9.6	2.5	1.2
S3	ND	ND	ND	ND	ND	ND	ND	4.4	ND	0.9
S4	ND	ND	ND	ND	ND	ND	ND	ND	ND	ND
S5	ND	ND	ND	ND	ND	ND	ND	ND	ND	ND

**Table 4 metabolites-13-00057-t004:** Concentration of SMZ and its metabolites in tissues of rat and pig.

Tissue	Time (Days)	Concentration of the Metabolites (μg/kg)		Time (Days)	Concentration of the Metabolites (μg/kg)	
		Rat			Pig	
		S0	S1		S0	S1
Liver	0.25	7730 ± 658	840 ± 102	0.5	1270 ± 345	2420 ± 582
	1	410 ± 76	72 ± 21	3	437 ± 58	124 ± 32
	3	210 ± 54	ND	7	200 ± 41	44 ± 17
	7	22 ± 8	ND	14	120 ± 33	ND
				28	22 ± 7	ND
Kidney	0.25	7140 ± 988	4046 ± 584	0.5	4355 ± 564	8645 ± 987
	1	379 ± 88	280 ± 79	3	503 ± 77	308 ± 56
	3	198 ± 65	70 ± 21	7	261 ± 43	93 ± 33
	7	118 ± 37	ND	14	125 ± 25	38 ± 11
Muscle	0.25	3657 ± 753	1833 ± 202	0.5	2391 ± 437	1890 ± 388
	1	204 ± 58	147 ± 54	3	365 ± 55	94 ± 33
	3	141 ± 36	80 ± 40	7	234 ± 48	30 ± 15
	7	58 ± 16	ND	14	90 ± 21	ND
Fat	0.25	1046 ± 194	798 ± 201	0.5	745 ± 102	1034 ± 321
	1	138 ± 32	52 ± 11	3	188 ± 54	65 ± 22
	3	75 ± 15	ND	7	98 ± 28	ND
	7	28 ± 9	ND	14	69 ± 16	ND

**Table 5 metabolites-13-00057-t005:** Concentration of SMZ and its metabolites in tissues of chickens.

Tissue	Time (Days)	Concentration of the Metabolites (μg/kg)				
		Chicken				
		S0	S1	S2	S3	S4
Liver	0.25	3622 ± 821	9159 ± 1148	ND	ND	ND
	3	1521 ± 423	184 ± 32	ND	ND	ND
	7	1013 ± 221	35 ± 8	ND	ND	ND
	14	365 ± 105	ND	ND	ND	ND
	28	41 ± 12	ND	ND	ND	ND
Kidney	0.25	20,951 ± 2198	234 ± 87	1058 ± 211	154 ± 45	108 ± 21
	3	1898 ± 478	89 ± 34	ND	ND	ND
	7	1012 ± 399	38 ± 11	ND	ND	ND
	14	521 ± 87	ND	ND	ND	ND
	28	22 ± 9	ND	ND	ND	ND
Muscle	0.25	5783 ± 1103	4209 ± 1897	ND	ND	ND
	3	1131 ± 325	769 ± 184	ND	ND	ND
	7	751 ± 98	443 ± 75	ND	ND	ND
	14	374 ± 56	134 ± 42	ND	ND	ND
	28	39 ± 8	ND	ND	ND	ND
Fat	0.25	480 ± 103	1120 ± 154	ND	ND	ND
	3	102 ± 33	76 ± 21	ND	ND	ND
	7	75 ± 19	ND	ND	ND	ND
	14	37 ± 13	ND	ND	ND	ND

**Table 6 metabolites-13-00057-t006:** Summary of the elimination parameters of the total radioactivity and major metabolites in the tissues of pigs, chickens and rats.

Species	Tissue	Compound	t_1/2_ (d)	Elimination Equation
Rat	Liver	Total Residues	1.87	C = 563·e^−kt^
		S0	1.39	C = 769·e^−kt^
		S1	0.22	C = 1648·e^−kt^
	Kidney	Total Residues	2.56	C = 795·e^−kt^
		S0	3.74	C = 769·e^−kt^
		S1	0.53	C = 2763·e^−kt^
	Muscle	Total Residues	2.41	C = 457·e^−kt^
		S0	3.47	C = 220·e^-kt^
		S1	0.72	C = 1094·e^−kt^
	Fat	Total Residues	2.36	C = 240·e^−kt^
		S0	2.63	C = 173·e^−kt^
Pig	Liver	Total Residues	5.76	C = 709·e^−kt^
		S0	5.63	C = 741·e^−kt^
		S1	1.19	C = 1820·e^−kt^
	Kidney	Total Residues	3.88	C = 2296·e^−kt^
		S0	4.29	C = 1477·e^−kt^
		S1	3.78	C = 445·e^−kt^
	Muscle	Total Residues	3.84	C = 1183·e^-kt^
		S0	4.09	C = 844·e^−kt^
		S1	1.15	C = 1449·e^−kt^
	Fat	Total Residues	5.80	C = 339·e^−kt^
		S0	5.81	C = 389·e^−kt^
Chicken	Liver	Total Residues	5.16	C = 3064·e^−kt^
		S0	4.90	C = 2697·e^−kt^
		S1	0.87	C = 5889·e^−kt^
	Kidney	Total Residues	4.59	C = 3542·e^−kt^
		S0	4.63	C = 3279·e^−kt^
		S1	2.61	C = 229·e^−kt^
	Muscle	Total Residues	5.16	C = 3234·e^−kt^
		S0	5.32	C = 1746·e^−kt^
		S1	3.64	C = 1478·e^−kt^
	Fat	total residues	5.36	C = 251·e^−kt^
		S0	5.29	C = 201·e^−kt^

No kinetic parameters were obtained because of the terminal phase was less than three time points. The elimination equation is described as follows: Ct = C_0_′e^−kt^, where Ct is the concentration at time t, C_0_′ is a pre-exponential term (fictitious concentration at t = 0) and k is the elimination rate constant. The half-life of elimination (t_1/2_) was calculated from the equation t_1/2_ = ln2/k.

## Data Availability

No new data were created or analyzed in this study. Data sharing is not applicable to this article.

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
