# Peer review of "Metabolic Disposition and Elimination of Tritum-Labeled Sulfamethoxazole in Pigs, Chickens and Rats"

_metabolites, 2022, doi:10.3390/metabo13010057_

Round 1
Reviewer 1 Report
The manuscript metabolites-2084265 entitled; “Comparative metabolism of tritum-labeled sulfamethoxazole in pigs, chickens and rats”. The authors investigate in a comparative study the metabolism of sulfamethoxazole antibiotic using rats, pigs, and chickens by radioactive tracing methods. Also, they determined residues of sulfamethoxazole in some tissues. In my opinion, the article has good data but a moderated presentation in some filled to cases poor presentation. Although the experiment is appropriately designed and implemented, there are minor grammar, stylistic, and syntax errors. In some cases, these errors negatively influence the understanding of the text. Also, there were no ethical approvement or statements please proved it.
General comments
- Please proofread the whole manuscript to avoid grammatical errors.
- In the abstract section, this section is not clear please rewrite it.
- In the introduction, a lot of statements without the corresponding citation, please revise the citations of all statements. Also, please use more recent related references.
- In the materials and methods section, please write the number of animals used in your study and the number of samples for each animal kind.
- Please describe all abbreviations in their first mention.
- In the discussion section, please be more specific, discuss your study with other similar studies and please state the superiorities of your study when compared to previous ones.
- In the references section, use journal style.
Author Response
请参阅附件。

Reviewer 2 Report
The reviewer pointed to several comments which may improve the quality of the manuscript before publishing. In addition, there are several important questions the reviewer pointed out need to be explained carefully.
In the Abstract section, please show the full names of “S1 and S0” as they first appear in the text.
Why were both rats and pigs administered by intramuscular injection and chickens by oral gavage?
Female and male rates were used for experiment, the reviewer wondering about the gender of pigs and chickens?
You said that six rats (three females and three males) were used for excretion and metabolism study. The reviewer wanted to know what the repetitions of the study were, whether the data have statistical significance is not clear.
Author Response
请参阅附件。

Reviewer 3 Report
General observation
The submitted manuscript describes the comparative metabolism, excretion and tissue distribution of sulfamethoxazole (SMZ) in pigs, chickens and rats. The design of study is suitable for the comparison of PK processes in different species and presentation of results is also appropriate. The authors have concluded that SMZ is rapidly eliminated from all three species and S1 is major metabolite of SMZ in all species. The radioactive residues of SMZ were of high concentration in blood, kidney and bile.
However, there are few observations/comments/suggestions which should be addressed during revision of this manuscript.
Specific comments
1. The title of manuscript does not reflects whole study presented in this manuscript as distribution of metabolites in different tissues and their excretion is equally described in this study.
2. Page 3, Line 113: The flow rate of mobile phase is mentioned as 10 mL/min. This seems to be very high flow rate. Please clarify if this was actual flow rate or a typo mistake?
3. Page 4, Line 162-165: the sentence is incomplete and vague.
4. In study design the species are mentioned but the number of animals is not mentioned. How many animals for each species were included in this study?
5. While male and female rats were included while there is no gender information is mentioned about pigs?
6. There are lot of contradictions in data presented in table 3 and description in text such as;
a) Line 200-201 states that the percentage of metabolites were compared with concentration of SMZ within 6 h after drug administration while in title of table 3 there is 12 h.
b) Lines 203-204 states that trace amount of S5 was detected in rat urine while in table this is mentioned as not detected (ND).
c) In table 1 male and female rats are mentioned separately while in table three there in no discrimination among rat in term of gender.
d) Line No 205 states that S0 and S1 were detected at 0-14 days while in title of table 3 there is mentioned that percentage of metabolites were compared within 12 hours of drug administration.
e) Lines 207 states that S0 & S1 were also detected in liver, kidney, muscles fats etc while no such information is presented in table 3.
7. Figure 3 should be places after table 3 for better understanding of the findings.
8. In summary, the authors used terms metabolism, excretion, distribution and elimination. In Pharmacokinetics, the term elimination is collectively used for metabolism and excretion. The statement should be revised as distribution and elimination.
9. In table 6 how fictitious concentration at t=0 was calculated? The equation presented can be used for prediction of drug concentration at any time after drug administration and this is not elimination equation. In my opinion the last column of table 6 is not necessary and the values should be replaced with elimination rate constant (K) instead of elimination equation.
10. The most important concern. How this study can help in safety evaluation and risk assessment of SMZ in food animals? A brief description may be provided in conclusion.
Round 2
Reviewer 1 Report
Thank you so much, all comments and suggestions are done as required.